# Clinical and Therapeutic Evaluation of the Ten Most Prevalent *CRB1* Mutations

**DOI:** 10.3390/biomedicines11020385

**Published:** 2023-01-27

**Authors:** Bruna Lopes da Costa, Masha Kolesnikova, Sarah R. Levi, Thiago Cabral, Stephen H. Tsang, Irene H. Maumenee, Peter M. J. Quinn

**Affiliations:** 1Department of Biomedical Engineering, Columbia University, New York, NY 10027, USA; 2Edward S. Harkness Eye Institute, Department of Ophthalmology, Columbia University Irving Medical Center/New York-Presbyterian Hospital, New York, NY 10032, USA; 3Jonas Children′s Vision Care, and Bernard & Shirlee Brown Glaucoma Laboratory, Department of Ophthalmology, Columbia University, New York, NY 10032, USA; 4Department of Ophthalmology, Federal University of São Paulo, São Paulo 04021-001, SP, Brazil; 5College of Medicine at the State University of New York at Downstate Medical Center, Brooklyn, NY 11203, USA; 6Vision Center Unit/EBSERH and Department of Ophthalmology, Federal University of Espírito Santo, Vitória 29075-910, ES, Brazil; 7Young Leadership Physicians Programme, National Academy of Medicine, Rio de Janeiro 20021-130, RJ, Brazil; 8Columbia Stem Cell Initiative, Columbia University, New York, NY 10032, USA; 9Department of Pathology & Cell Biology, Columbia University, New York, NY 10032, USA

**Keywords:** prime editing, base editing, CRISPR, *Crumbs homolog 1* (*CRB1*), inherited retinal disease (IRD), maculopathy, retinitis pigmentosa, Leber Congenital Amaurosis (LCA)

## Abstract

Mutations in the *Crumbs homolog 1* (*CRB1*) gene lead to severe inherited retinal dystrophies (IRDs), accounting for nearly 80,000 cases worldwide. To date, there is no therapeutic option for patients suffering from *CRB1*-IRDs. Therefore, it is of great interest to evaluate gene editing strategies capable of correcting *CRB1* mutations. A retrospective chart review was conducted on ten patients demonstrating one or two of the top ten most prevalent *CRB1* mutations and receiving care at Columbia University Irving Medical Center, New York, NY, USA. Patient phenotypes were consistent with previously published data for individual *CRB1* mutations. To identify the optimal gene editing strategy for these ten mutations, base and prime editing designs were evaluated. For base editing, we adopted the use of a near-PAMless Cas9 (SpRY Cas9), whereas for prime editing, we evaluated the canonical NGG and NGA prime editors. We demonstrate that for the correction of c.2843G>A, p.(Cys948Tyr), the most prevalent *CRB1* mutation, base editing has the potential to generate harmful bystanders. Prime editing, however, avoids these bystanders, highlighting its future potential to halt *CRB1*-mediated disease progression. Additional studies investigating prime editing for *CRB1*-IRDs are needed, as well as a thorough analysis of prime editing’s application, efficiency, and safety in the retina.

## 1. Introduction

Mutations in the *Crumbs homolog 1* (*CRB1*) gene cause severe and disabling inherited retinal dystrophies (IRDs). There are approximately 80,000 *CRB1* patients worldwide, with a prevalence in the United States of 1 in 86,500 [1,2,3]. *CRB1* is associated with various phenotypes including Leber Congenital Amaurosis type 8 (LCA8), retinitis pigmentosa type 12 (RP12), and maculopathy [4,5,6]. The age of disease onset varies and defines the subtype. LCA8 can be diagnosed in the first few months of life and is associated with nystagmus [2]. Typically, LCA8 presents with severe vision loss as demonstrated with electroretinography (ERG) that reveals non-recordable photopic and scotopic responses. RP12, on the other hand, presents in adolescence with night blindness and midperipheral scotomas. ERG demonstrates attenuated scotopic responses followed by a decrease in photopic responses. Vision loss is progressive, with severe visual impairment and blindness presenting by the fourth decade of life [2].

The fundus findings of *CRB1*-associated IRDs include an atrophic macula, preservation of the para-arteriolar retinal pigment epithelium (PPRPE), and Coats disease [4,5,7]. Suggestive findings on optical coherence tomography (OCT) imaging include a thickened retina, in contrast to the typically thinned retina of other IRDs, and a lack of laminated retinal layers [6]. Genotype–phenotype correlation studies have been largely unsuccessful. However, nonsense mutations and frameshift deletions are associated with increased disease severity, and Motta et al. found a direct correlation between mutation effect and patient phenotype [7,8,9,10].

The CRB complex consists of the CRB protein family (CRB1, CRB2, and CRB3), the PALS1-PATJ-MUPP1 protein complex, and the PAR6-PAR3-aPKC-CDC42 protein complex [11,12,13]. The CRB complex regulates apical–basal polarity, apical membrane size and promotes maintenance of cell adhesion via adherens junctions [14,15,16,17]. The *CRB1* gene is located on the long arm of chromosome 1 (1q31.1), spanning 210 kb of genomic DNA [4,18]. The canonical CRB1 isoform, CRB1-A, codes for a transmembrane protein localized to the sub-apical region of photoreceptors (PRs) and Müller glial cells (MGCs) in the human retina [4,19,20,21,22,23,24]. While several CRB1 isoforms exist, only three are significantly expressed in the human retina: CRB1-A, CRB1-B, and CRB1-C [13,25]. In mice, CRB1-A and CRB1-B operate predominantly in different cell types, MGCs and PRs, respectively [25]. In the mouse retina, loss of CRB1 and/or CRB2 leads to retinal degeneration mimicking RP and LCA [20,25,26,27,28,29,30,31,32,33]. In *CRB1*-RP patient induced pluripotent stem cell (iPSC)-derived retinal organoids, the corresponding phenotype includes disruptions of the outer limiting membrane (OLM) and ectopic PRs above the OLM [24]. This model’s phenotype is similar to that seen in mice lacking CRB1 or expressing the C249W CRB1 variant [20,34].

There is currently no intervention available for *CRB1*-associated IRDs. Due to the various CRB1 isoforms and their predominant cell-type-specific localizations, a therapeutic approach such as gene supplementation is complicated [25]. Recent advances have demonstrated the success of adeno-associated virus (AAV)-mediated gene augmentation of *CRB2* in *Crb1Crb2* conditional knockout mice [23]. However, *CRB1-A*-mediated gene augmentation provides only mild morphological benefits without functional rescue. Further, it is associated with several adverse effects potentially due to ectopic or overexpression of human CRB1 proteins in *Crb* mouse models [23,35]. Moreover, *CRB1* mutant rats with a severe early-onset retinal phenotype have not responded to *CRB1-A* or *CRB2* AAV-mediated gene augmentation [36]. These data highlight the need for a CRB1 isoform-independent treatment modality, such as precise genome editing technology applying clustered regularly interspaced short palindromic repeats (CRISPR).

Early generations of CRISPR relied on the RNA-guided CRISPR-Cas nuclease system. Upon recognition and binding between the protospacer and Cas-RNA complex, the catalytic domains of the Cas endonuclease induce double-stranded breaks (DSBs), which are then repaired via eukaryotic host cell non-homologous end joining (NHEJ) or homology-directed repair (HDR) [37,38]. NHEJ results in arbitrary insertions and deletions in the target zone, leading to frameshift mutations, premature stop codons, and gene knockout [37]. Alternatively, host cells can engage in HDR to generate precise DNA modifications at the target region with the aid of an exogenous template [39]. However, HDR is limited to actively dividing cells, an obstacle to successful editing in post-mitotic cells such as MGCs and PRs [39]. 

To overcome the limitations of the RNA-guided CRISPR-Cas nuclease system, base editing was created. There are two well-developed base editing strategies—cytosine base editors (CBE), which create C-G to T-A transitions, and adenine base editors (ABE), which create A-T to G-C changes. Base editors allow for single-strand DNA modification through specific, single nucleotide alterations without creating DSBs [40,41]. In the past, base editing has been primarily limited by its inability to perform transversions. Recent studies have shown the promise of glycosylase base editors to perform C-to-A and C-to-G transversions [42,43,44]. However, base editing still has limitations such as (1) bystanders due to more than one target nucleotide within the editing window and (2) its inability to perform all 12 base-to-base changes [45]. 

To overcome the shortcomings of base editing, prime editing was created. With its ability to re-write DNA sequences using a prime editing guide RNA (pegRNA) in conjunction with a prime editor consisting of the H840A Streptococcus pyogenes Cas9 (spCas9) nickase linked to Moloney murine leukemia virus (MMLV) reverse transcriptase (RT). Prime editing can perform targeted insertions, deletions, and all 12 types of point mutations, in a DSB independent manner [46]. Although promising for the future of gene therapy, prime editing is not always the preferential therapy over base editing. Base editing offers a higher editing efficiency the majority of the time and a lower rate of indel formation [46].

Together, base editing and prime editing are potential alternatives to gene augmentation for the treatment of *CRB1*-IRDs, particularly in light of CRB1 retinal isoform diversity [25,47,48]. This study evaluates base editing and prime editing strategies for the ten most frequent variants in *CRB1,* estimated to account for 31.62% of all reported alleles (Table 1). We find that base editing is suitable for six of the ten *CRB1* mutations. However, bystander consideration can significantly affect the applicability of base editing as a therapeutic approach to treat *CRB1*-associated IRDs. For example, for the most prevalent *CRB1* mutation, c.2843G>A p.(Cys948Tyr), base editing is not the optimal approach due to a series of bystander mutations that can potentially affect gene function. On the other hand, prime editing is feasible for all ten *CRB1* mutations, showing it to be a flexible tool for the treatment of *CRB1*-linked IRDs. Further, we provide a clinical evaluation based both on our cohort of patients and the literature for these prevalent *CRB1* mutations.

## 2. Materials and Methods

### 2.1. Patient Selection

The ten most frequent *CRB1* variants, as per the LOVD database, were cross-referenced with 24 patients seen and evaluated at the Edward S. Harkness Eye Institute at Columbia University Irving Medical Center (New York, NY, USA) with a clinical diagnosis of inherited retinal dystrophy and confirmatory genetic testing, showing two pathogenic mutations in the CRB1 gene. Ten patients with at least one of the ten most frequent variants were included for evaluation. This study was conducted under Columbia University Institutional Review Board-approved protocol IRB AAAF1849, and all procedures were performed according to the tenets of the Declaration of Helsinki. Informed consent was waived due to the retrospective nature of this study and the minimal risk conferred to patients as described in Columbia University Institutional Review Board-approved protocol AAAR8743.

### 2.2. Clinical Evaluation

All patients underwent a complete ophthalmic examination and imaging; six patients had full-field electroretinography (ffERG) testing. Each patient’s best corrected visual acuity was measured before dilation using topical tropicamide (1%) and phenylephrine hydrochloride (2.5%). Patients underwent functional imaging including SD-OCT (Spectralis HRA2, Heidelberg Engineering, Heidelberg, Germany), SW-AF (Spectralis HRA2, Heidelberg Engineering, Heidelberg, Germany), and wide-angle color fundus photography using an Optos 200Tx unit (Optos; PLC, Dunfermline, UK). Full-field electroretinography testing was performed using a Diagnosys Espion Electrophysiology System (Diagnosys LCC, Littleton, MA, USA). 

### 2.3. Base Editing Analysis

The LOVD database was searched for previously reported variants in *CRB1* (https://databases.lovd.nl/shared/variants/CRB1/unique; accessed on 10 October 2022). The top ten most commonly reported variants were selected for review (Table 1). The variants were analyzed for suitable base editing and prime editing approach for correction using SnapGene (Software version 4.3.11, GSL Biotech, Boston, MA, USA). Each variant was analyzed by at least two different individuals based on the outlined criteria below.

ABEs perform A to G transitions while CBEs install C to T transitions. Of the ten CRB1 mutations, five are suitable for ABE and one is suitable for CBE. None of the transversions are correctable by the most recent glycosylase base editor. The transition mutations were evaluated for base editing feasibility using the near-PAMless SpCas9 variant SpRY, as previously described [49]. The editing window included nucleotides at positions 4–7 of the protospacer for ABEs and at positions 4–8 for CBEs [50]. Designs in the top DNA strand are labeled as positive (+) and those in the bottom strand are labeled as negative (−). The sequence immediately downstream to the protospacer was recorded as the PAM for each position from 4–7 (CBE) or 4–8 (ABE) and the editing window was evaluated for bystanders. The effect of each bystander on the amino acid sequence and larger protein sequence was recorded and evaluated in terms of potential effects in gene function (Table 3). 

### 2.4. Prime Editing Analysis

The top ten most prevalent *CRB1* mutations described in the LOVD database were evaluated for correction using prime editing using SnapGene (Software version 4.3.11, GSL Biotech, Boston, MA, USA). Each variant was analyzed by at least two different individuals based on the outlined criteria below. Although prime editors using near-PAMless SpCas9 variant SpRY are already available, prime editing has a low PAM displacement dependency [46]. As such, here, we only considered the use of both the canonical NGG PAM prime editor and the NGA PAM prime editor. For those mutations with multiple possible designs, we chose the NGG or NGA design that led to a nick closest to the mutation. Designs in the top DNA strand are labeled as positive (+) and those in the bottom strand are labeled as negative (−). Designs up to ten nucleotides from the nick were included in the evaluation. Designs were made using only pegRNA; no nicking sgRNA designs were considered. Specific protospacers were identified for each mutation from which 3′-extensions were designed. Prime editing efficiencies are highly associated with the composition of the pegRNAs, as such optimizations of primer binding sequences (PBS) and RT template are necessary to achieve the best prime editing activity [46]. Here, we chose a PBS of thirteen nucleotides in length and a RT template of sixteen nucleotides in length, not accounting for insertions and deletions that require a longer RT [46,50].

## 3. Results and Discussion

### 3.1. Clinical Results

A total of 10 patients with at least one confirmed mutation in the ten *CRB1* variants selected from LOVD (Table 1) were evaluated. Within this cohort of patients, five were heterozygous and five were homozygous for mutations in *CRB1*. One patient was excluded from analysis as no clinical history was available. The age at onset of symptoms ranged from birth to 12 years, with a mean of 5.5 years old. Best corrected visual acuity ranged from 0.5 to 2.8, with a mean logarithm of the minimum angle of resolution (LogMAR) score of 1.59. Eight patients underwent more than one visit. With a range of 2–18 visits, the mean follow-up time for patients with more than one visit was 3.75 years. Six patients underwent ERG testing, showing largely extinguished responses with minimal preservation of photopic relative to scotopic responses (data not shown). The demographic and genetic information are summarized in Table 2.

**Table 1 biomedicines-11-00385-t001:** The 10 Most Frequent *CRB1* Variants in the Leiden Open Variation Database.

Variant #	cDNA Change	Protein Change	Exon	Reported Alleles (n)	Proportion of Alleles (%)
**1**	c.2843G>A	p.(Cys948Tyr)	9	260	12.48
**2**	c.2401A>T	p.(Lys801*)	7	73	3.50
**3**	c.2234C>T	p.(Thr745Met)	7	72	3.45
**4**	c.2290C>T	p.(Arg764Cys)	7	64	3.07
**5**	c.2688T>A	p.(Cys896*)	8	43	2.06
**6**	c.498_506del	p.(Ile167_Gly169del)	2	42	2.02
**7**	c.613_619del	p.(Ile205Aspfs*13)	2	34	1.63
**8**	c.1576C>T	p.(Arg526*)	6	30	1.44
**9**	c.3307G>A	p.(Gly1103Arg)	9	21	1.01
**10**	c.614T>C	p.(Ile205Thr)	2	20	0.96
**Total**				659	31.62

*Optos* color fundus photography revealed extensive intraretinal pigment migration in five of nine patients, with relative para-arteriolar sparing (Figure 1A–E). The three patients under the age of ten years revealed little to no pigment migration (Figure 1G–I). Patient 6 revealed a central area of atrophy and mild subretinal and intraretinal pigment migration within the macula only (Figure 1F). Short-wavelength autofluorescence (SW-AF) imaging revealed severely diminished autofluorescence in all but three patients. Patients 3 and 7 revealed similar patterns on SW-AF: extensive 360-degree hypoautofluorescence with patches of atrophy throughout the macula and the peripheral retina (Figure 2A,B). Patient 6 revealed homogeneous diminished autofluorescence within the macula, extending nasotemporally past the arcades (Figure 2C). Spectral domain optical coherence tomography (SD-OCT) imaging revealed a consistent phenotype with poorly laminated retinal layers, foveal hypoplasia, and increased retinal thickness in all patients (Figure 2D). The average retinal thickness among these patients was 395 microns (normal = 212 ± 20 microns) [51]. Patient 8 revealed small cystoid macular edema (CME) nasal to the fovea.

### 3.2. Evaluation of Clinical Results

Of the nine patients evaluated, eight presented with classic signs and symptoms of *CRB1*-associated LCA. These patients had four of the top ten variants: c.2401A>T;p.Lys801Ter, c.2843G>A;p.Cys948Tyr, c.3307G>A;p.Gly1103Arg, and c.2234C>T p.Thr745Met. Age of disease onset was between birth and childhood with SD-OCT imaging demonstrating diffuse disorganization of retinal layers and increased retinal thickness [6]. For those patients with peripheral changes, pigment migration patterns were consistent with the literature, demonstrating preservation of the para-arteriolar retinal pigment epithelium [4]. Only patient 6, with the variants c.2843G>A;p.Cys948Tyr and c.498_506delAATTGATGG; p.Ile167_Gly169del presented with an alternate phenotype. The mutation at c.498_506del has previously been associated with isolated maculopathy, as opposed to LCA or RP [52,53]. According to the literature, these patients have a later age at onset and have more preserved visual acuities [52]. Patient 6 has a genotype and phenotype consistent with the published data. 

### 3.3. Therapeutic Development for CRB1-Linked Inherited Retinal Dystrophies

For a multitude of IRDs, researchers have extensively explored therapeutic modalities, including gene therapy, cell transplantation, optogenetics, and retinal prosthesis [54]. These therapeutic modalities hope to slow or halt disease progression and/or, in part, restore lost vision. Gene therapies, gene augmentation and gene editing, are suitable for early-stage intervention, where there is still some preservation of photoreceptors. However, cell transplantation and optogenetics are more appropriate as mid to late-stage therapeutic options, where there is considerable photoreceptor degeneration and vision loss. At these advanced stages, it is necessary to replace the photoreceptor cells (cell transplantation) or modulate the expression of light-sensitive molecules (optogenetics) in surviving cell types of the retina enabling those cells to compensate for the photoreceptors light response. Lastly, retinal prosthesis can restore a basic sense of sight and are for patients exhibiting profound vision loss [55,56,57]. However, which therapeutic modality best suits a patient is reflected in their current stage of disease progression [13]. The natural history of a disease will dictate how quickly a patient may have to move from one potential therapeutic modality to the next and highlight the optimal timing of an intervention during disease progression. 

Currently, there is no therapeutic option for patients suffering from *CRB1*-linked IRDs. Mutations in *CRB1* are associated with various phenotypes, including LCA8, RP12, and maculopathy [1,2,3]. For several years, no clear genotype–phenotype correlations were found for *CRB1*-linked IRDs, but null mutations were more associated with increased disease severity. However, several studies have now shown that the c.498_506del, p.(Ile167_Gly169del) *CRB1* mutation is associated with *CRB1*-linked maculopathy and other correlations may now be found when reevaluating *CRB1* mutations in light of CRB1 isoform diversity [2,7,8,9,10,55,56]. Natural history studies, in particular prospective studies, are essential in determining the therapeutic window of opportunity, patient eligibility criteria, and clinical endpoints of future trials assessing treatment efficacy [57]. Recently, several retrospective and prospective studies have begun to evaluate the optimal therapeutic window for *CRB1*-linked IRDs [2,56,58,59,60]. In *CRB1*-linked LCA/ early-onset severe retinal dystrophy (EOSRD) patients will likely need to receive therapeutic intervention within the first decade of life due to early functional and structural involvement of the macular. However, patients with *CRB1*-linked RP and maculopathy have a more significant therapeutic window spanning the first three decades of life [2,57,58,59,60,61]. Therefore, early interventional therapies, such as gene augmentation and gene editing, may prove efficacious for patients with *CRB1*-linked IRDs.

As previously discussed in the introduction, gene augmentation may not be the optimal approach to treat *CRB1* patients due to the complexity of CRB1 isoform diversity [25]. However, the novel approach of using CRB1 family member CRB2 is being evaluated and has shown proof-of-concept in *Crb1Crb2* conditional knockout mice [23]. Further, the possibility of using CRB1-B alone or concomitant expression of CRB1-A and CRB1-B is being explored [62,63]. Here, we prioritize the evaluation of DSB-independent gene editing modalities base editing and prime editing as early interventional strategies for correcting *CRB1* mutations. 

Effective delivery methods for base and prime editing in vivo are still being developed. Due to the large size of base editors (~6 kb) and prime editors (~7 kb) (including corresponding sgRNA and pegRNA, respectively) their in vivo delivery can be a limitation that significantly impacts the efficiency of these techniques [45,64]. Currently, adeno-associated viruses (AAV) are the “go to” delivery vehicle for retinal gene therapy but they only have a packaging capacity of ~4.7 kb, making them unsuitable as a single delivery vehicle for the current generation of editors [45]. To date, split-intein *trans*-splicing approaches using dual AAV vectors for the delivery of split base and prime editors have been developed and successfully delivered in vivo to the retina and retinal pigment epithelium (RPE) [65,66,67,68,69]. Both base and prime editing have been succesfuly utilized to recover the functional deficit, as measured by electrinorgaphy (ERG), in the retinal degeneration 12 (*rd12*) mouse model, a representative model of human with *RPE65* mutations [66,68,70]. In their initial publication, the Palczewski Lab used lentiviral-mediated delivery of an ABE and a sgRNA achieving an average editing efficiency of 15.95% and as high as 29% [70]. In a subsequent follow-up publication, they found they could further improve editing efficiency (average editing efficiency 22% and as high as 57%) by testing additional ABE variants with different protospacer-adjacent motif (PAM) sequences. Importantly, as lentivirus are associated with safety concerns they tested the feasibility of a dual AAV split base editor staregy, finding a low base editing efficiency ~2.7%. Using a prime editing approach in the same rd12 mouse model specifically using a triple AAV-mediated delivery method (split prime editor between two AAVs and a third AAV carrying the pegRNA) Jang et al. found they could achieve an average prime editing efficiency of 6.4% [68]. All methodologies found minimal indels or off-target mutations [66,68,70]. However, the use of non-mammalian inteins previously were considered a potential safety concern due the persistence as *trans*-splicing byproducts. Nevertheless, inclusion of a degron has been utilized in the *trans*-splicing system allowing for rapid ubiquitination and proteasomal degradation of these byproducts increasing their safety profile [71]. Optionally, untethered prime editing in which the Cas9 and reverse transcriptase are not linked, can be delivered in separate AAVs without using *trans*-splicing approaches [49]. Dual approaches have allowed the required increase in cargo size but are typically less efficient than single vector delivery [64,72]. Optimization of smaller base and prime editors that fit within in the packaging constraints of a single AAV are being explored and are a priority of viral methods to be moved forward with. Recently, Davis et al. evaluated a series of ABE8e variants that use compact CjCas9, Nme2Cas9 and SauriCas9 targeting domains to develop a collection of single-AAV high-activity base editors [73]. 

Despite these on going advances in viral vector-mediated delivery approaches for base and prime editors, viral-mediated delivery suffers from additional constraints other then that of packaging limits. These include anti-capsid immunity and vector-induced immunogenicity, genotoxicity, therapeutic potency, and high production costs [74]. Therefore, non-viral delivery methods hold great promise with their ability to circumvent many of these issues including that of packging constraints. A comprehensive overview of non-viral delivery modalities is well reviewed by Salman and colleagues, covering lipid based, polymeric and inorganic nanoparticle delivery methods [74]. Non-viral particles have flexible biophysical properties, making them suitable for the delivery of hydrophilic and hydrophobic molecules, as well as proteins and CRISPR/Cas components. Importantly, they can have their surface functionalized with special ligands to improve cell specificity and therefore, safety. Recently Herrera-Barrera, Ryals and colleagues developed peptide-guided lipid nanoparticles for the delivery of mRNA to the neural retina of rodents and nonhuman primates [75]. Toxicity and low neuroretina transfection efficiency are still potential limitations in the use of non-viral vectors for the treatment of retinal diseases, but the chance of changing the nanoparticles composition and size offers possibilities to decrease toxicity and achieve desired safety, cargo capacity and transfection efficiency.

#### 3.3.1. Evaluation of Base Editing

In the human retina, three CRB1 isoforms—CRB1-A, CRB1-B and CRB1-C—are expressed [25]. The development of a gene therapy approach that could correct all three CRB1 isoforms would be advantageous. Genetic mutations that change amino acid sequences affecting CRB1-A, CRB1-B and CRB1-C isoforms, may lead to negative effects on protein function. Therefore, the evaluation of the potential effects of bystanders is crucial to determine if the target mutation is suitable for base editing correction. Of the six common variants that are transition mutations, only one is highly amenable to base editing due to its total lack of bystanders within the editing window: the c.614T>C variant (Table 3, Figure 3). There are three other variants—c.1576C>T, c.2234C>T, and c.3307G>A—that have at least one base editing design without any bystanders. Two variants, c.2290C>T and c.3307G>A, have base editing designs with bystanders that lead to silent mutations that can be classified as negligible. However, the way we should see synonymous variants in health, disease, and protein folding is a current point of debate [76]. These five variants are all potentially base editable and have been marked in green in Table 3.

**Table 3 biomedicines-11-00385-t003:** Base editing design for the six transition mutations within the top 10 most frequent *CRB1* variants in the Leiden Open Variation Database.

	Mutation	Strand	Position	Spacer (5′ to 3′)	PAM	Bystanders (BS)	BS Effects
**ABE**	c.1576C>T	−	4	TTCaGAAAAGTAGAAGAGCC	ATT	BS1 and BS2	BS1 + BS2 = Phe525Pro BS1 alone = Phe525Ser (isoform B = 152) (isoform A&C = 525)
			5	CTTCaGAAAAGTAGAAGAGC	CAT	BS1	
			6	GCTTCaGAAAAGTAGAAGAG	CCA	N/A	
			7	TGCTTCaGAAAAGTAGAAGA	GCC	N/A	
	c.2234C>T	−	4	AGCaTTCGGACAAACATGGA	GAG	N/A	Leu746Pro (isoform A) Leu373Pro (isoform B)
			5	AAGCaTTCGGACAAACATGG	AGA	N/A	
			6	GAAGCaTTCGGACAAACATG	GAG	N/A	
			7	TGAAGCaTTCGGACAAACAT	GGA	BS1	
	c.2290C>T	−	4	CACaGATATATTGATAAGTG	CTG	BS2	BS1 is a silent; BS2 Ile763Thr (isoform A) Ile390Thr (isoform B)
			5	ACACaGATATATTGATAAGT	GCT	BS2	
			6	GACACaGATATATTGATAAG	TGC	BS1	
			7	AGACACaGATATATTGATAA	GTG	BS1	
	c.2843G>A	+	4	TAGaTATTGCAAATGCTGTT	TTT	BS2	BS1 is intronic (−2); BS2 Ile949Val (isoform A) Ile5676Val (isoform B)
			5	TTAGaTATTGCAAATGCTGT	TTT	BS2	
			6	ATTAGaTATTGCAAATGCTG	TTT	BS1	
			7	CATTAGaTATTGCAAATGCT	GTT	BS1	
	c.3307G>A	+	4	ATCaGAGGCATTTATCTCTC	TTA	BS2	BS2 is silent; BS1 Ile1102Val (isoform A) Ile729Val (isoform B)
			5	AATCaGAGGCATTTATCTCT	CTT	BS2	
			6	AAATCaGAGGCATTTATCTC	TCT	N/A	
			7	GAAATCaGAGGCATTTATCT	CTC	BS1	
**CBE**	c.614T>C	+	4	AAAcAGGAAGATATACTTGT	ATC	N/A	N/A
			5	GAAAcAGGAAGATATACTTG	TAT		
			6	TGAAAcAGGAAGATATACTT	GTA		
			7	ATGAAAcAGGAAGATATACT	TGT		
			8	AATGAAAcAGGAAGATATAC	TTG		

Green text represents base editing designs without bystanders or with bystander effects that lead to silent mutations.

The base editing designs that generate bystanders associated with amino acid changes should be avoided since they can potentially affect protein function. For example, the bystanders created by some of the base editing designs for the c.1576C>T variant produce an amino acid change from phenylalanine to proline or serine. Phenylalanine is an aromatic amino acid, whereas both proline and serine are non-aromatic [50,77,78]. The molecular structure of amino acids is closely related with their physical and chemical properties. As such, changing amino acids with different molecular structures are likely to significantly affect protein conformation and function. 

For the most prevalent *CRB1* mutation, c.2843G>A, all base editing designs lead to bystanders, that can potentially affect the gene function. For two of the possible designs (target nucleotide at positions 4 and 5 within the editing window) we have the bystander BS2 that leads to a change from isoleucine to valine. This change is preferable given the many similarities between isoleucine and valine, as both are aliphatic and hydrophobic [79,80]. However, isoleucine has one carbon and two hydrogens more than valine which may affect protein structure and conformation; further protein modeling and functional analysis are required to definitively predict the effects of this variant on protein function. For the other two designs (target nucleotide at positions 6 and 7 within the editing window), we have the intronic bystander BS1 at position -2. Nucleotides positioned at exon–intron boundaries are frequently associated with the consensus “cis” sequences that are recognized by splicing machinery. Therefore, changes in nucleotides that are in splicing sites can lead to the incorrect intron removal and thus cause improper protein translation [81,82]. Together, BS1 and BS2 illustrate that base editing might not be the ideal gene editing approach to correct the most prevalent *CRB1* mutation. 

Base editing is an ever-evolving gene editing technique. In 2020, Walton et al. created a near-PAMless Cas9 variant that greatly increased the base editing capabilities currently available [38]. This near-PAMless Cas9 was used in our study of the most common *CRB1* variants as it allowed for the least restricted analysis of base editing potential. We were able to evaluate each of the top ten variants for base editing capability only analyzing the presence of bystanders within the editing window and we have demonstrated that base editing can serve as therapeutic approach for five of the ten most common *CRB1* variants. While SpRY Cas9 approach increases the possibility of base editing, it is frequently associated with lower editing efficiencies when compared to the wild-type Sp Cas9 [83]. SpRY Cas9 is also associated with higher off-targeting rates, which may compromise the safety of this approach when applying the technique in vivo [84]. Additionally, SpRY Cas9 has been associated with high levels of plasmid integration, making delivery methods such as AAV unsuitable, further limiting its use [84]. Alternatively, use of the conventional Cas9 base editor could increase the efficiency and may decrease off-targeting, however, the wild-type Cas9 is highly limited by the PAM displacement. In our evaluation, base editing with conventional Sp Cas9 is not suitable for any of the top ten *CRB1* mutations, considering both (1) the lack of an ideally positioned NGG PAM and (2) the creation of bystanders. Therefore, the near-PAMless Cas9 would allow the use of base editing to correct five of the ten most frequent *CRB1* variants. Off-targeting and delivery method must be further evaluated to confirm the feasibility of base editing for each specific mutation.

#### 3.3.2. Evaluation of Prime Editing 

Prime editing, contrary to base editing, can install all types of transitions, transversions, insertions, and deletions, making this CRISPR-based technology suitable for the correction of more than 70% of the known human pathogenic genetic variants [46]. Of the top ten *CRB1* mutations, six are transitions, two are transversions, and two are deletions. All of those have suitable prime editing designs using either NGG or NGA PAM designs (Table 4, Figure 4). Anzalone et al. demonstrated that prime editing has a low dependency of PAM displacement. Mutations positioned far away from the PAM (at position ±33, for example) are still targetable by prime editing and can give rise to efficient editing [46]. Here, as an example, we highlight prime editing strategies with the edit positions close to the PAM. However, there is no clear correlation with edit position and editing efficiency [46]. All the top ten *CRB1* mutations are theoretically prime editable.

Prime editors can precisely target the mutation and write the desired sequence that will be incorporated in the final edited strand. Therefore, only the specific edit will be installed, rendering prime editing free of bystanders. However, common to all CRISPR-based technology, the off-targeting ratios are variable and must be analyzed for each mutation individually. Researchers often identify potential off-target sites and evaluate them using methods including next generation sequencing (NGS), nickase-based Digenome sequencing (nDigenome seq) and circulization for in vitro reporting of cleavage effects by sequencing (CIRCLE-seq). However, it is important to perform genome-wide off-target evaluations as these methodologies often underestimate off-targeting rates [45]. Additionally, to date, we still do not fully understand potential cell-type-specific differences for on and off-targeting with prime editing in the retina. Further, there have been concerns regarding the potential read through of the RT beyond the pegRNAs 3’ extension sequence (PBS and RTT) into the scaffold. Nonetheless, prime editing requires three DNA hybridization steps (complementarity of the target DNA to each component of the pegRNA: spacer, PBS, and RT template) that substantially reduce its off-targeting effects compared to editing methods based only on the hybridization between target DNA and sgRNA [46]. In addition, here, we are not proposing the use of a near-PAMless (SpRY) prime editor, which has been previously described to be frequently associated with higher off-targeting ratios [84].

As with every gene editing technology, prime editing has its caveats, which include the optimization of the length of PBS and RT template that make up the 3′ extension of the pegRNA [46]. Further, type 3 polymerase III promoters such as U6, 7SK, and H1 that are frequently used for the expression of guide RNAs and pegRNAs, require a T-stretch as a termination signal for transcription. Gao et al. showed that T4 (TTTT) works as a minimal terminator and that six or more consecutive T nucleotides are required for full termination efficiency [85]. Three designs (labeled with double asterisks in Table 4) have a T-stretch of T4 or more, which could affect the prime editing efficiency due to the generation of fewer full-length pegRNAs in addition to truncated and dysfunctional pegRNAs that would compete with any full-length pegRNAs. However, this limitation is more problematic when delivering the prime editing machinery as plasmids DNA, which requires further transcription inside of the cells. Alternatively, in vitro mRNA synthesis mediated by T7 RNA polymerases can overcome this issue due to its more restricted termination signals [86,87,88]. Therefore, delivering the pegRNA as mRNAs can overcome the T-stretch limitation of type 3 polymerases. Additionally, mRNA seems to be less toxic to cells and it has been the preferential delivery method for prime editing in vivo [88]. Taken together, this information demonstrates that prime editing is a promising CRISPR-based technology to provide an isoform-independent therapeutic approach to treat the ten most prevalent *CRB1* mutations.

## 4. Conclusions

In this study, we present the evaluation of the top ten most prevalent variants in the *CRB1* gene for feasibility of base and prime editing. Of the six transition mutations in this list amenable to base editing, only the c.614T>C; p.Ile205Thr variant is free of bystanders, making it an attractive *CRB1* mutation to develop an initial base editing therapeutic around. On the other hand, the *CRB1* variants showed great promise for prime editing as all ten variants had possible prime editing designs. Further evaluation revealed that only seven of these designs might be feasible due to consecutive T nucleotides in the design sequence. In particular, the development of a prime editing therapeutic initially for the c.2843G>A; p.(Cys948Tyr) *CRB1* mutation would be attractive owing to its prevalence in the patient population and the presence of bystander mutations complicating a base editing approach for this mutation. 

The main limitation of this study is that it is in silico in nature. Therefore, it only highlights the potential amenability of these gene editing technologies to a particular variant but lacks experimental data on the performance and efficacy of the base and prime editing systems at the in silico evaluated loci. Several important considerations should be taken into account including chromatin accessibility effects on efficacy, cell type-specific DNA repair mechanisms, off-targeting, on-target indels, bystanders and efficient delivery of editing components to the cell type of choice. 

This proof-of-concept evaluation of base and prime editing shows the potential promise for the clinical application of gene editing for *CRB1* mutations. We must now move forward with the systematic screening of these potential *CRB1* base and prime editing therapeutics using in vitro knockin cell lines and patient iPSC-derived retinal organoids. 

## Figures and Tables

**Figure 1 biomedicines-11-00385-f001:**
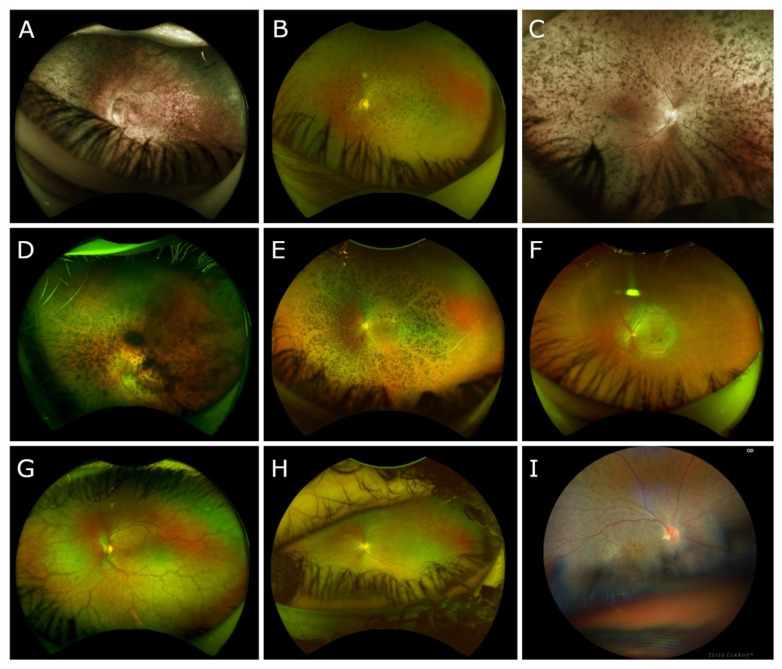
Color fundus imaging for nine patients with at least one confirmed mutation among the ten most prevalent mutations in *CRB1*. (**A**–**E**) Patients 1, 3, 4, 5, and 7, respectively, with ages ranging from 20 to 49 years old, demonstrated extensive intraretinal pigment migration 360-degrees throughout the periphery with relative para-arteriolar sparing. (**F**) Patient 6 presented with significant macular atrophy extending beyond the arcades with central pigmentary changes. There is no peripheral pigment migration. (**G**–**I**) Patients 2, 8, and 9, respectively, who are under the age of 10, demonstrated rare pigment migration scattered throughout the posterior pole and peripheral retina.

**Figure 2 biomedicines-11-00385-f002:**
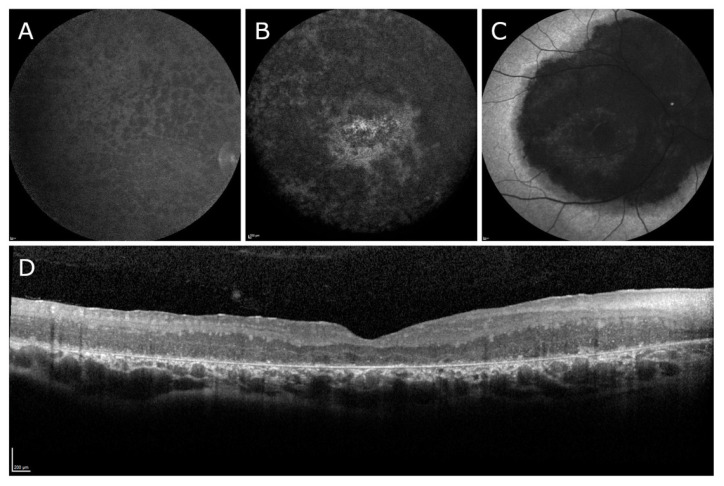
Short-wavelength autofluorescence (SW-AF) for three patients with at least one confirmed mutation in the top ten most prevalent variants of *CRB1* and representative spectral domain optical coherence topography (SD-OCT) imaging of patients with two mutations in *CRB1*. (**A**,**B**) Two SW-AF images from patients 3 and 7, respectively, representative of typical SW-AF presentation in patients with confirmed *CRB1* mutations. There is globally diminished autofluorescence with overlying heterogeneous pattern of hypoautofluorescence involving the posterior pole and peripheral retina. (**C**) SW-AF of patient 6 showing central atrophy in the macula extending nasally past the optic nerve, superiorly greater than inferiorly. (**D**) SD-OCT imaging from patient 2 representative of typical SD-OCT presentation in patients with confirmed *CRB1* mutations. Retinal layers are poorly delaminated. There is foveal hypoplasia and increased retinal thickness.

**Figure 3 biomedicines-11-00385-f003:**
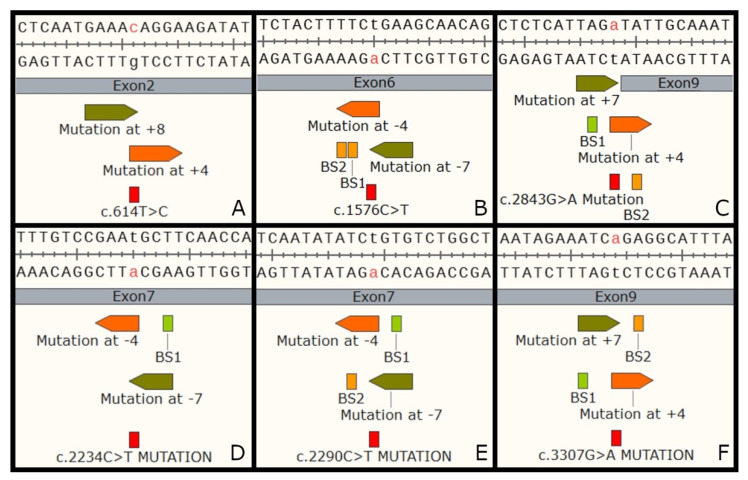
Base editing designs for the six trasition variants—c.614T>C (**A**), c.1576C>T (**B**), c.2843G>A (**C**), c.2234C>T (**D**), c.2290C>T (**E**), c.3307G>A (**F**)—in the top 10 most frequent *CRB1* mutations.

**Figure 4 biomedicines-11-00385-f004:**
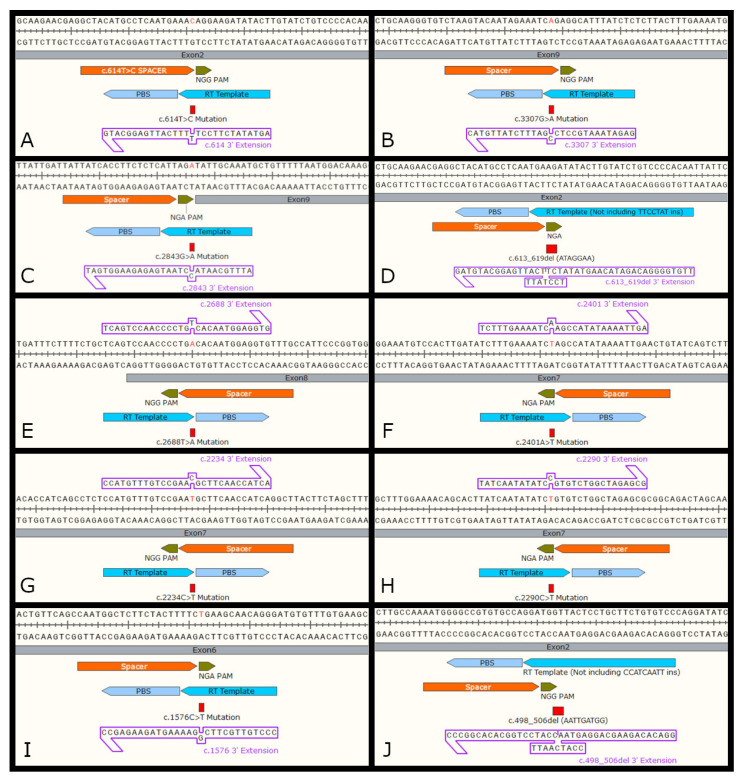
Prime editing design for the top 10 prevalent *CRB1* mutations—c.614T>C (**A**), c.3307A>G (**B**), c.2843AG>A (**C**), c.613_619del (ATAGGAA)(**D**), c.2688T>A (**E**), c.2401A>T (**F**), c.2234C>T (**G**), c.2290C>T (**H**), c.1576C>T (**I**), and c.498_506del (AATTGATGG) (**J**).

**Table 2 biomedicines-11-00385-t002:** Demographic, genetic, and clinical summary of patients.

Patient ID	1	2	3	4	5	6	7	8	9
Age and sex	44F	8M	26M	24M	49F	40M	20F	9M	5M
Age at onset *	childhood	3	birth	4	childhood	12	4	8	2
BCVA (LogMar) OD, OS	2.8, 2.4	0.9, 0.9	1.9, 1.9	2.4, 2.4	2.4, 2.4	0.9, 1.3	1.6, 1.5	0.5, 0.5	1.1, 0.9
Follow-up (years)	1	5	9	2	n/a	8	1	1	3
Number of visits	2	4	6	2	1	17	3	7	8
Retinal thickness (microns)	423	489	437	327	n/a	382	292	296	515
Genetic variant 1 ^‡^	c.2401A>T p.Lys801Ter	c.2843G>A p.Cys948Tyr	c.2843G>A p.Cys948Tyr	c.3307G>A p.Gly1103Arg	c.2234C>T p.Thr745Met	c.2843G>A p.Cys948Tyr	c.3307G>A p.Gly1103Arg	c.3307G>A p.Gly1103Arg	c.3307G>A p.Gly1103Arg
Genetic variant 2	c.2706C>G p.Cys902Trp	c.2480G>T p.Gly827Val	c.2245_2247del p.749delSer	homozygous	c.257_258dup p.Asn87*	c.498_506delAATTGATGG; p.Ile167_Gly169del	homozygous	homozygous	homozygous

* Age at onset of symptoms as noticed by the patient or a parent/guardian. Not equivalent to age at diagnosis. ^‡^ Genetic variant 1 represent those in the ten most frequent *CRB1* variants.

**Table 4 biomedicines-11-00385-t004:** Prime editing design for the top 10 most frequent *CRB1* variants in the Leiden Open Variation Database.

Mutation	PAM	Edit Position	Spacer	3′ Extension
c.2843G>A	AGA	+6	TATTATCACCTTCTCTCATT	ATTTGCAATACCTAATGAGAGAAGGTGAT
c.2234C>T	CGG	−1	AGCCTGATGGTTGAAGCATT	CCATGTTTGTCCGAACGCTTCAACCATCA
c.2401A>T	AGA	−4	CAGTTCAATTTTATATGGCT **	TCTTTGAAAATCAAGCCATATAAAATTGA
c.2290C>T	AGA	−4	GCCGCGCTCTAGCCAGACAC	TATCAATATATCCGTGTCTGGCTAGAGCG
c.2688T>A	GGG	−1	CAAACACCTCCATTGTGACA	TCAGTCCAACCCCTGTCACAATGGAGGTG
c.613_619del	AGA	+4	GAGGCTACATGCCTCAATGA	TTGTGGGGACAGATACAAGTATATCTTCCTATTTCATTGAGGCATGTAG
c.498_506del	TGG	+6	AATGGGGCCGTGTGCCAGGA	GGACACAGAAGCAGGAGTAACCATCAATTCCATCCTGGCACACGGCCC
c.1576C>T	TGA	+4	CAATGGCTCTTCTACTTTTC **	CCCTGTTGCTTCGGAAAAGTAGAAGAGCC
c.3307G>A	AGG	+2	CTAAGTACAATAGAAATCAG	GAGATAAATGCCTCCGATTTCTATTGTAC
c.614T>C	AGG	+3	GCTACATGCCTCAATGAAAC	AGTATATCTTCCTTTTTCATTGAGGCATG**

** Designs with a T-stretch of T4 or more, which could affect editing efficiency. Green text represents edit to be installed. Red represents mutation.

## Data Availability

All data generated or analyzed during this study are included in this published article.

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
