# Peer review of "Clinical and Therapeutic Evaluation of the Ten Most Prevalent CRB1 Mutations"

_biomedicines, 2023, doi:10.3390/biomedicines11020385_

Round 1

Reviewer 1 Report

The authors present a clinical observation and in-silico analysis of nine documented CRB1 coding variants and their corresponding clinical features in carriers of inherited retinal dystrophies. They also evaluate the feasibility and potential challenges of using base editing or prime editing methods to correct these variants. The manuscript offers valuable insights into the translation of genetic editing strategies into clinical therapies. However, some areas could be improved before publication.

Major comments

1. The authors should be advised to provide a ranked list of CRB1 mutations/variants from their own study population. These data will significantly contribute to the existing data aggregation, such as the LOVD database. The reported data suggested that the author's study population could have a unique CRB1 mutation profile.

2. The authors should include more detailed information about the specific tools and software used in the design and in-silico analysis of the base editing or prime editing systems. This would provide valuable context for readers and allow for a better understanding and replication of the results.

3. The authors should consider discussing the different administration routes of gene therapies, such as intravitreal injection, subretinal injection, delivery via viral vectors, and delivery via lipid particles, in the context of base editing or prime editing systems on inherited retinal diseases. This would provide valuable insight into the potential clinical applications and practical considerations of these treatments.

4. The authors should discuss the limitations of their manuscript, including the lack of experimental data about the performance and efficacy of the proposed base editing or prime editing systems. This would provide a more balanced perspective on the potential of these therapies and help to highlight areas for further research.

Minor points

1. The authors should carefully proofread their manuscript to correct any minor typographical or grammatical errors. This will improve the clarity and readability of the text and enhance the overall quality of the work. E.g., "...a clinical di-agnosis of...", "...the ten most fre-quent variants...", "and all proce-dures were".

2. PBS was not spelled out when it first appeared.

3. Table 2: data from which eyes were presented for the BCVA (LogMar) ?

Author Response

We thank the reviewer for their enthusiasm for our manuscript and their helpful comments that have improved it. Please see comments addressed below.

Reviewer 1

Major comments

The authors should be advised to provide a ranked list of CRB1 mutations/variants from their own study population. These data will significantly contribute to the existing data aggregation, such as the LOVD database. The reported data suggested that the author's study population could have a unique CRB1 mutation profile.

Response: Table 2 provides information on the patients evaluated within this study and lists the CRB1 mutations associated with each patient. Here, we are evaluating nine CRB1 patients and 4 CRB1 mutations. However, we analyzed base editing and prime editing for the top 10 CRB1 mutations to expand the concept of the paper and provide a more complete and relevant analysis.

The authors should include more detailed information about the specific tools and software used in the design and in-silico analysis of the base editing or prime editing systems. This would provide valuable context for readers and allow for a better understanding and replication of the results.

Response: We have further clarified our description of base and prime editing analysis. We have provide a detailed description of how we performed base “2.3 Base Editing Analysis” and prime editing “2.4 Prime Editing Analysis” evaluations as well as provided Tables of relevant spacer sequences, PAMs, bystanders and their effects (Table 3 and Table 4). Further, annotations of snap gene files with relevant information for base (Figure 3) or prime editing (Figure 4) designs of each mutation are provided. There are several online tools, such as Prime Design (drugthatgene.pinellolab.partners.org) and CRISPR RGEN (rgenome.net), that help researchers to design potential base and prime editing strategies for specific mutations. However, those tools do not take into consideration some limitations as the presence of multiple T in the gRNAs sequences. Here all designs were done manually by at least two different individuals, this is clarified in the material and methods section.

The authors should consider discussing the different administration routes of gene therapies, such as intravitreal injection, subretinal injection, delivery via viral vectors, and delivery via lipid particles, in the context of base editing or prime editing systems on inherited retinal diseases. This would provide valuable insight into the potential clinical applications and practical considerations of these treatments.

Response: We thank the reviewer for this suggestion. We have addressed these points by adding a paragraph to section 3.3.

The authors should discuss the limitations of their manuscript, including the lack of experimental data about the performance and efficacy of the proposed base editing or prime editing systems. This would provide a more balanced perspective on the potential of these therapies and help to highlight areas for further research.

Response: We thank the reviewer for this important addition to the manuscript. We have addressed these limitations in section 5. Conclusions.

Minor points

The authors should carefully proofread their manuscript to correct any minor typographical or grammatical errors. This will improve the clarity and readability of the text and enhance the overall quality of the work. E.g., "...a clinical di-agnosis of...", "...the ten most fre-quent variants...", "and all proce-dures were".

Response: We thank the reviewer for bringing this formatting issue to our attention; we have gone through and corrected these minor typographical or grammatical errors.

PBS was not spelled out when it first appeared.

Response: Corrected.

Table 2: data from which eyes were presented for the BCVA (LogMar) ?

Response: Thanks for drawing this unclarity to our attention. The first number is OD and second number is OS. We have edited the able to make this clearer.

Reviewer 2 Report

My comments

1, table 1: It is better to provide the exon positions where the mutation sites are located.

2, Table 4: Of the ten mutations, which one would you like to try first for the gene therapy? Why?

3, Off-target is of great concern in the gene therapies.  The researcher have identified fewer off-target in the prime editing technique. Is it possible to evaluate any potential off-target mutations through whole-genomes based on the sequences of the primer binding site (PBS)?

Author Response

We thank the reviewer for their helpful comments that have improved our manuscript. Please see comments addressed below.

Reviewer 2

Table 1: It is better to provide the exon positions where the mutation sites are located.

Response: We thank the reviewer for the comment. We have amended table 1 as suggested.

Table 4: Of the ten mutations, which one would you like to try first for the gene therapy? Why?

Response: We thank the reviewer for this comment. We would suggest to develop a prime editing strategy for the c.2843G>A, p.(Cys948Tyr) mutation initially due to its prevalence in the patient population and the fact that our analysis showed several bystanders being present making a base editing strategy potentially more problematic. We have emphasized this point in the section 5. Conclusions.

Off-target is of great concern in the gene therapies. The researcher have identified fewer off-target in the prime editing technique. Is it possible to evaluate any potential off-target mutations through whole-genomes based on the sequences of the primer binding site (PBS)?

Response: We thank the reviewer for the comment. This is an important comment, and we have addressed it further in section 3.3.2.

Round 2

Reviewer 1 Report

The authors have addressed my concerns. Thanks for the revision.